# Trend analysis and projection of the gastric cancer disease burden in Taiwan during 1990–2021: An analysis of the global burden of disease study 2021

Xiaohuang Yang 1,2☯, Shaoxing Chen3☯, Canmei Zhong1, Yadong Lai2*, Fenglin Chen1*

1 Department of Gastroenterology and Fujian Institute of Digestive Disease, Fujian Medical University Union Hospital, Fuzhou, Fujian, China, 2 Department of Gastroenterology, Zhangzhou Affiliated Hospital of Fujian Medical University, Zhangzhou, Fujian, China, 3 Department of Radiation Oncology, Zhangzhou Affiliated Hospital of Fujian Medical University, Zhangzhou, Fujian, China

☯ These authors contributed equally to this work.
* drchenflxiehe@163.com (FC); lyd0596@hotmail.com (YL)

## Abstract

### Background

Gastric cancer is a serious health threat to people in Taiwan. This study reports gastric cancer burden and trends in Taiwan, from 1990 to 2021, and projects the incidence and mortality trends through 2036.

### Methods

Data on gastric cancer burden in Taiwan came from the 2021 GBD database. Trend changes were evaluated via joinpoint analysis, with the Age-Period-Cohort model estimating age, period, and cohort influences. Decomposition analysis measured contributions of population growth, aging, and epidemiological factors, and the BAPC model forecast future burden.

### Results

Between 1990 and 2021, the age-standardized incidence rate (ASIR), age-standardized prevalence rate (ASPR), age-standardized mortality rate (ASMR), and age-standardized disability-adjusted life-year (DALY) rate (ASDR) of gastric cancer in Taiwan all showed a decreasing trend. In terms of absolute burden, the number of incident cases in 2021 was 4,839 (95% uncertainty interval [UI]: 4,272–5,352), representing a 41.20% increase, prevalent cases reached 9,835 (95% UI: 8,664–11,032), a 60.36% rise, deaths totaled 3,799 (95% UI: 3,345–4,220), up by 29.17%, and DALYs were 79,425 (95% UI: 71,168–87,813), with a 0.19% decrease. A significant gender disparity was observed, with males bearing a heavier burden. Projections indicate that incident cases and deaths will continue to rise through 2036.

**Data availability statement:** The data used in this study are publicly available from the GBD 2021 database, accessible at http://ghdx.healthdata.org/gbd-results-tool.

**Funding:** This work was supported by grants to the Young and Middle-aged Talents Training Project of Fujian Provincial Health Technology Project (2021GGA021); Startup Fund for Scientific Research, Fujian Medical University (2020QH1098); Joint Funds for the National Key Clinical Specialty Construction Projects of Fujian Province, China (Grant No. 2023 − 1594) and Minimally Invasive Medical Center of Fujian, China (Grant No. [2017]171). The Young and Middle-aged Talents Training Project of Fujian Provincial Health Technology Project (2021GGA021), which played a role in the decision to publish. Additionally, the Startup Fund for Scientific Research Fujian Medical University (2020QH1098), Joint Funds for the National Key Clinical Specialty Construction Projects of Fujian Province, China (Grant No. 2023−1594) and Minimally Invasive Medical Center of Fujian, China (Grant No. [2017]171) provided support for this work, contributing to data collection and analysis.

**Competing interests:** The authors have declared that no competing interests exist.

## Conclusions

Across the past 32 years, gastric cancer in Taiwan has exhibited a declining relative burden, yet the absolute burden has risen consistently and is forecast to keep increasing over the next 15 years. Precision risk control measures and context-specific public health policies are required to alleviate this burden.

## Introduction

Gastric cancer is a common malignant tumor, during the first half of the 20th century, it was the leading cause of cancer-related deaths in Europe and the United States [1]. GLOBOCAN 2022 stats indicate that gastric cancer makes up 4.9% of all new cancer cases worldwide, with 968,350 people diagnosed each year, ranking fifth. It's also the fifth leading cause of cancer deaths, with 659,853 deaths in 2022 [2]. Data from China's National Cancer Center shows that gastric cancer was the fifth most incident and third most fatal malignancy in China in 2022, accounting for 358,700 new cases and 260,400 deaths [3]. According to 2018 data from the Taiwan Cancer Registry, gastric cancer mortality ranked sixth among all cancers in both sexes in Taiwan, underscoring its critical role in the regional cancer profile, potentially related to local environment, lifestyle, and genetics [4].

While the exact causes of gastric cancer are not fully established, epidemiological research is key to clarifying its etiology. Identified risk factors associated with gastric cancer development include Helicobacter pylori (HP) infection, smoking, alcohol use, poor lifestyle choices, low fruit and vegetable consumption, gastroesophageal reflux disease, and familial predisposition [5–10]. Gastric cancer incidence differs by gender, and while there's no clear reason why, some theories suggest female sex hormones may offer protection [11,12]. Regional variations in gastric cancer burden have been marked over recent decades, while age-standardized incidence has declined globally—a primary epidemiological characteristic [5,13]. This trend is strongly associated with improved food storage, evolving dietary practices, and successful HP eradication. These factors act on multiple stages of gastric cancer initiation and progression, reducing the risk of incidence to some extent and thereby shaping the global epidemiological pattern of declining gastric cancer rates. Against this backdrop, advancing in-depth epidemiological studies on gastric cancer is of great significance. Such research will not only enhance understanding of its epidemiological trends and underlying determinants across populations and geographic areas but also inform the design of targeted, personalized gastric cancer control strategies, offering substantial guidance and support for global public health initiatives in gastric cancer prevention.

There is a large body of Global Burden of Disease (GBD) database-based research on gastric cancer burden, predominantly covering global populations [14–17] and with a substantial focus on mainland China [18,19]. Notably, no studies have reported on the gastric cancer burden in Taiwan. Furthermore, research has shown that advanced gastric cancer resulted in an economic burden of $423 million in Taiwan in 2013, representing 0.08% of the region's economic aggregate [20].

Grasping the epidemiological trends of gastric cancer in Taiwan is essential to reducing economic burden, developing preventive strategies, and enhancing population health. This study used the latest 2021 GBD data to fully evaluate gastric cancer burden measures in Taiwan and predict how cases and deaths will trend over the next 15 years. Our results should help improve understanding of the burden of gastric cancer, assess how well current prevention efforts are working, and provide information to create better, more scientific prevention measures for the disease in Taiwan.

## Methods

### Data source

Led by the Institute for Health Metrics and Evaluation (IHME), the 2021 GBD database covers 204 countries and territories globally, documenting health effects from 371 diseases, injuries, and 88 risk factors [21,22]. For this study, gastric cancer burden data spanning 1990–2021 for mainland China and Taiwan were retrieved using the Global Health Data Exchange (GHDx) query tool (https://vizhub.healthdata.org/gbd-results/).GBD employs complex modeling approaches, including DisMod-MR and spatiotemporal Gaussian process regression (ST-GPR), for estimating cancer burden data [23].In the GBD framework, 95% uncertainty intervals (UIs) are employed to compute estimates of gastric cancer burden. Uncertainty intervals for all indicators are calculated using mean estimates from 1,000 iterations, where the 95% UI is defined by the 2.5th and 97.5th percentiles of the distribution [21].

### Joinpoint regression analysis

Temporal trends in gastric cancer burden in Taiwan were assessed via Joinpoint analysis, which identifies inflection points marking significant changes in trends [24–26]. The approach partitions trends into several phases, computing the Annual percentage change (APC) and 95% Confidence Interval (CI) for each phase, before using the Average annual percentage change (AAPC) to synthesize overall trend variations between 1990 and 2021. When both the AAPC and the lower end of its 95% CI are above 0, the trend is rising in that specific period. If both the AAPC and the upper end of its 95% CI are below 0, the trend is falling during that time [27].

**Age–period–cohort analysis.** An age–period–cohort model was used to analyze age, period, and birth cohort trends in gastric cancer incidence and mortality in Taiwan from 1990 to 2021. This model enables investigation of how these factors influence gastric cancer burden across three temporal dimensions: age, period, and birth cohort [28]. However, the age–period–cohort model is subject to collinearity, a complete linear dependency among age, cohort, and period (cohort=period-age), which prevents traditional regression models from accurately estimating the independent effects of age, period, and cohort. In this study, the intrinsic estimator algorithm is used to eliminate collinearity via linear constraints on the parameter space, thereby producing unique, stable estimates with minimal bias [29]. The specific steps to derive the intrinsic estimator are described in a leading reference [30]. Given the specific characteristics of the intrinsic estimator, we stratified ages into 17 groups (15–19 years to ≥95 years) at 5-year intervals, alongside 22 birth cohorts (1897–1901 to 2002–2006) and 6 periods (1992–1996 to 2017–2021), to present gastric cancer incidence and mortality for specific ages, periods, and birth cohorts.

**Decomposition analysis.** Using Das Gupta's decomposition approach, this study quantified the contributions of population growth, aging, and epidemiological shifts to gastric cancer incidence, prevalence, mortality, and disability-adjusted life-years (DALYs) in Taiwan between 1990 and 2021 [31,32]. Such a method allows for detailed evaluation of how each factor drives disease burden, providing a basis for formulating targeted strategies.

**Bayesian age-period-cohort (BAPC) analysis.** A BAPC model was utilized to forecast gastric cancer incidence and mortality burden in Taiwan between 2022 and 2036. Using integrated nested Laplace approximation (INLA) for Bayesian inference, the model captures temporal trends across age, period, and cohort, and predicts future shifts in disease burden using historical data and model estimates [33]. The model's basic equation is given by:

$$\log(Rate_{ijt}) = \alpha + \mu_i + \beta_j + \gamma_k + \varepsilon_{ijt}$$

Rate$_{ijt}$ represents the incidence or mortality rate at time t, for age group j, and cohort k. α is the intercept term. μ$_i$ represents the period effect that varies over time i.β$_j$ represents the age effect that varies across age group j. γ$_k$ represents the cohort effect that varies across birth cohort k.ε$_{ijt}$ is the error term, capturing unobserved variation or random fluctuations.

**Data visualization.** All analyses and data visualization were performed using the R tool 4.4.2 and Joinpoint software (version 5.2.0). Statistical significance was defined as $P < 0.05$.

**Ethical considerations.** All data in this study were sourced from the publicly accessible GBD database and subjected to secondary analysis, rendering ethical approval and informed consent unnecessary.

## Results

### Description of the disease burden of gastric cancer in Taiwan

Between 1990 and 2021, gastric cancer in Taiwan saw a declining relative burden but a rising absolute burden. Incident cases grew from 3,427 (95% UI: 2,983–3,756) in 1990 to 4,839 (95% UI: 4,272–5,352) in 2021, representing an overall increase of 41.20%. The age-standardized incidence rate (ASIR) fell from 21.46 (95% UI: 18.65–23.57) per 100,000 in 1990 to 11.51 (95% UI: 10.17–12.71) per 100,000 in 2021. The number of prevalent cases increased from 6,133 (95% UI: 5,325–6,759) in 1990 to 9,835 (95% UI: 8,664–11,032) in 2021, with a cumulative increase of 60.36%. Deaths rose from 2,941 (95% UI: 2,552–3,219) in 1990 to 3,799 (95% UI: 3,345–4,220) in 2021, an increase of 29.17%. The age-standardized mortality rate (ASMR) decreased from 19.35 (95% UI: 16.76–21.15) per 100,000 in 1990 to 8.86 (95% UI: 7.84–9.81) per 100,000 in 2021.DALYs went from 79,573 (95% UI: 69,145–87,025) in 1990 to 79,425 (95% UI: 71,168–87,813) in 2021. Men had a significantly higher burden of gastric cancer than women. At the same time, the trend in gastric cancer burden in mainland China was similar to that in Taiwan (Table 1).

### Gender disparities in gastric cancer burden by age group in Taiwan in 2021

For gastric cancer in Taiwan in 2021, incident cases in the 15–19 years age group were equal between males and females, but in all other age groups, males had more cases. The male incidence peak was in the 65–69 years group, whereas the female peak was in the 80–84 years group (Fig 1A). Prevalent cases were higher in males across all age groups, with both sexes peaking in the 65–69 years group (Fig 1C). Most age groups had more deaths in males than females, with male deaths peaking at 65–69 years and female deaths at 80–84 years (Fig 1E). Females had lower Disability-Adjusted Life Years (DALYs) than males across all age groups, with 60–69 years showing markedly higher DALYs in both sexes relative to other age groups (Fig 1G). Across all age groups, female crude incidence rates (CIR), crude prevalence rates (CPR), crude mortality rates (CMR), and crude DALY rates (CDR) were lower than those in males, with a consistent upward trend observed with increasing age. CPR peaked at 80–84 years in males and 90–94 years in females, the other three indicators peaked at 90–94 years in both sexes (Fig 1B, 1D, 1F, 1H).

### Age differences in gastric cancer burden in Taiwan, 1990 and 2021

In 1990, the number of new gastric cancer cases in Taiwan was higher in age groups younger than 45 years than in 2021, the opposite was true for age groups 45 years and older. Notably, in age groups 80 years and older, new cases in 2021 were significantly higher than in 1990, potentially attributable to population aging and increased life expectancy (Fig 2A). Prevalent cases in 1990 surpassed those in 2021 among age groups under 40 years, whereas the reverse was true for groups aged 40 years and older (Fig 2B). In terms of deaths and DALYs, 1990 figures were higher than 2021 figures in age groups under 75 years (Fig 2C, 2D). CIR, CPR, CMR, and CDR in 2021 were higher than in 1990 for the 85–89 and 90–94 years age groups, all other age groups had lower rates in 2021 (Fig 2).

**Table 1. Gastric cancer cases and Age-Standardized Rates in Taiwan and the Chinese mainland for 1990 and 2021.**

| Location | Measure | Sex | 1990 | | 2021 | |
|---|---|---|---|---|---|---|
| | | | All-ages cases | Age-standardized rates per 100,000 people | All-ages cases | Age-standardized rates per 100,000 people |
| | | | n (95% UI) | n (95% UI) | n (95% UI) | n (95% UI) |
| Taiwan | Incidence | Both | 3427 (2983-3756) | 21.46 (18.65-23.57) | 4839 (4272-5352) | 11.51 (10.17-12.71) |
| Taiwan | Incidence | Male | 2469 (2046-2806) | 29.55 (24.46-33.39) | 3264 (2853-3695) | 16.77 (14.65-18.91) |
| Taiwan | Incidence | Female | 958 (892-1025) | 12.73 (11.77-13.61) | 1575 (1349-1756) | 6.99 (6.06-7.76) |
| Taiwan | Prevalence | Both | 6133 (5325-6759) | 35.63 (30.80-39.34) | 9835 (8664-11032) | 23.94 (21.16-26.81) |
| Taiwan | Prevalence | Male | 4499 (3697-5132) | 49.42 (40.60-56.46) | 6909 (5922-7993) | 35.57 (30.67-41.01) |
| Taiwan | Prevalence | Female | 1635 (1526-1755) | 20.16 (18.79-21.62) | 2927 (2554-3252) | 13.59 (11.91-15.04) |
| Taiwan | Deaths | Both | 2941 (2552-3219) | 19.35 (16.76-21.15) | 3799 (3345-4220) | 8.86 (7.84-9.81) |
| Taiwan | Deaths | Male | 2102 (1725-2378) | 26.70 (22.08-30.12) | 2507 (2181-2827) | 12.85 (11.19-14.43) |
| Taiwan | Deaths | Female | 839 (779-899) | 11.67 (10.78-12.52) | 1292 (1092-1466) | 5.54 (4.74-6.25) |
| Taiwan | DALYs | Both | 79573 (69145-87025) | 469.63 (407.34-515.02) | 79425 (71168-87813) | 193.70 (174.50-213.30) |
| Taiwan | DALYs | Male | 56199 (46145-63576) | 631.96 (519.42-714.41) | 53778 (47031-60988) | 277.60 (243.37-314.26) |
| Taiwan | DALYs | Female | 23374 (21973-24956) | 289.33 (270.46-308.73) | 25647 (22552-28495) | 120.17 (106.45-132.71) |
| China | Incidence | Both | 407471 (337565-477569) | 48.03 (40.22-56.69) | 611799 (471966-765562) | 29.05 (22.42-36.20) |
| China | Incidence | Male | 278596 (208188-346574) | 67.64 (51.71-83.67) | 446434 (325932-589284) | 44.48 (32.18-58.38) |
| China | Incidence | Female | 128875 (105159-157699) | 30.23 (24.80-36.89) | 165365 (127716-208140) | 15.23 (11.77-19.16) |
| China | Prevalence | Both | 615217 (503482-720422) | 67.17 (55.35-78.41) | 1226056 (943897-1546818) | 57.23 (44.18-71.99) |
| China | Prevalence | Male | 431491 (319789-537802) | 94.77 (70.95-117.86) | 937643 (683807-1240047) | 89.25 (65.15-117.49) |
| China | Prevalence | Female | 183726 (148406-226732) | 40.33 (32.86-49.41) | 288412 (223730-365255) | 26.71 (20.68-33.89) |
| China | Deaths | Both | 374066 (310921-442251) | 46.05 (38.88-54.43) | 445013 (344736-555834) | 21.51 (16.66-26.61) |
| China | Deaths | Male | 251602 (188204-314409) | 64.67 (49.60-79.93) | 314779 (230725-418722) | 32.61 (23.61-42.80) |
| China | Deaths | Female | 122464 (100701-149718) | 29.81 (24.69-36.43) | 130234 (100509-163561) | 12.02 (9.29-15.10) |
| China | DALYs | Both | 10773457 (8850977-12638919) | 1181.61 (978.38-1390.90) | 10642127 (8222106-13383779) | 501.26 (387.29-627.98) |
| China | DALYs | Male | 7399279 (5450273-9252517) | 1634.85 (1218.61-2045.07) | 7740359 (5634331-10365104) | 750.39 (550.90-997.91) |
| China | DALYs | Female | 3374178 (2734453-4160646) | 743.14 (605.54-913.28) | 2901768 (2251657-3679391) | 268.83 (208.91-340.98) |

### Trends of age-specific burden of gastric cancer in Taiwan from 1990 to 2021

Between 1990 and 2021, males in Taiwan had higher numbers of incident, prevalent, and fatal gastric cancer cases, and higher DALYs, than females. In both sexes, those aged 65 years and older represented over 50% of cases. Over time, the disease burden in older age groups increased steadily (Fig 3). The data showed a slow rise in incident cases in both males and females, while fatal cases declined marginally over the period (Fig 3A, 3C).

### Joinpoint analysis of gastric cancer burden in Taiwan, 1990–2021

Joinpoint analysis was used to examine trends and sex disparities in gastric cancer indicators in Taiwan between 1990 and 2021. ASIR and Age-standardized prevalence rate (ASPR) trended upward initially before declining between 1990 and 2021, with peak values in 1997 (Fig 4A, 4B). ASMR and Age-standardized DALY rate (ASDR) followed a pattern of decline, then increase, then further decline, with peaks observed in 1997 (Fig 4C, 4D). Stratification by sex showed that male age-standardized rates (ASRs) were consistently higher than female ASRs during the 32-year period (Fig 4). From 1990 to 2021, the APCC for age-standardized incidence decreased by 1.99% overall, 1.71% in males, and 1.96% in females. For age-standardized mortality, APCC declined by 2.4% in both males and females, and both sexes decreased of 2.58% (S1 and S2 Tables).

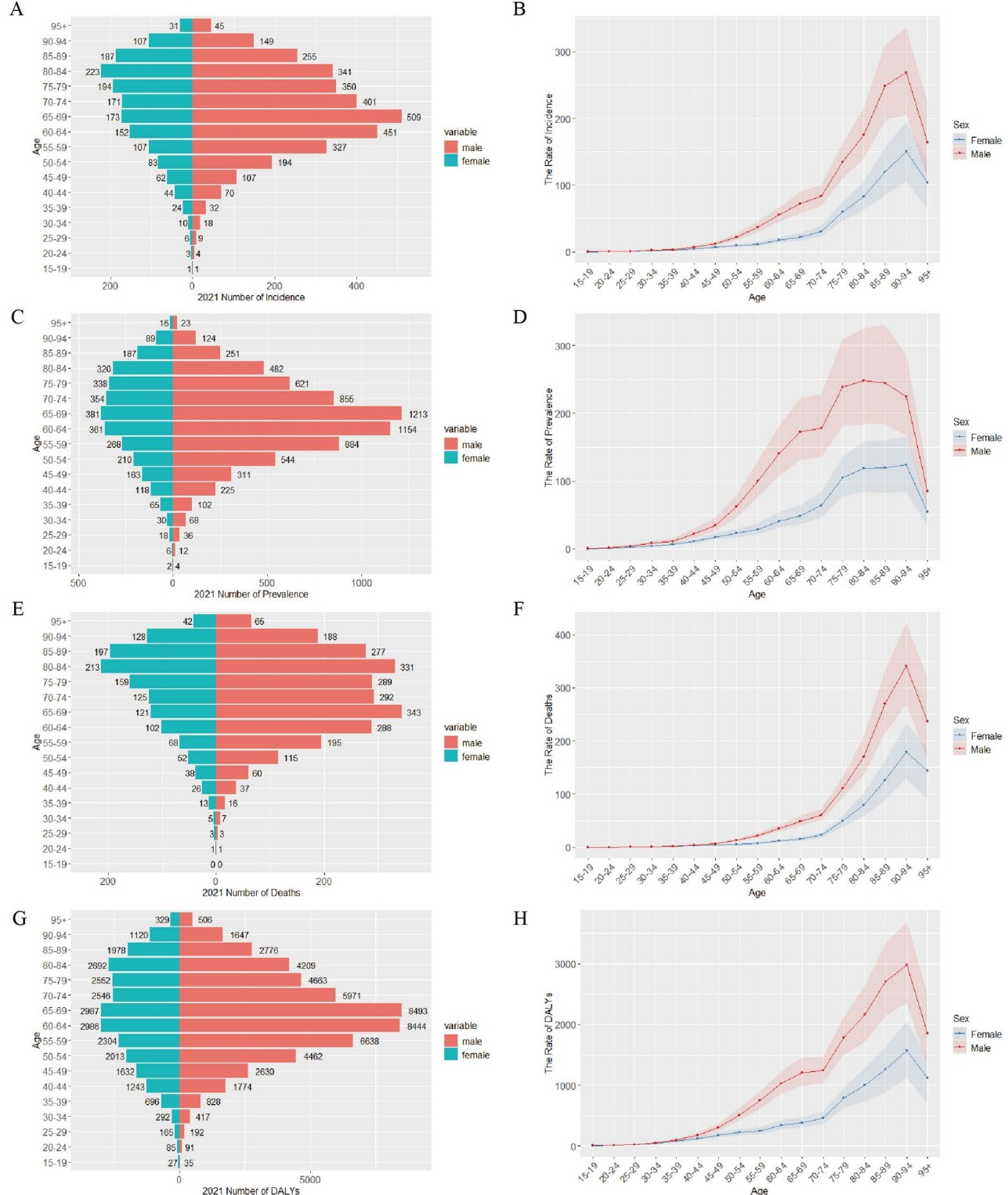

**Fig 1. Age-specific numbers and rates of incidence, prevalence, deaths and DALYs for gastric cancer in Taiwan, 2021.** (A) The number of incident cases of gastric cancer by age and sex. (B) CIR of gastric cancer by sex. (C) The number of prevalent cases of gastric cancer by age and sex. (D) CPR of gastric cancer by sex. (E)The number of deaths due to gastric cancer by age and sex. (F) CMR of gastric cancer by sex. (G)The number of

DALYs due to gastric cancer by age and sex. (H) CDR of gastric cancer by sex. Abbreviations: CIR, Crude Incidence Rate; CMR, Crude Mortality Rate; CPR, Crude Prevalence Rate; CDR, Crude DALYs Rate; DALYs, Disability-Adjusted Life Years.

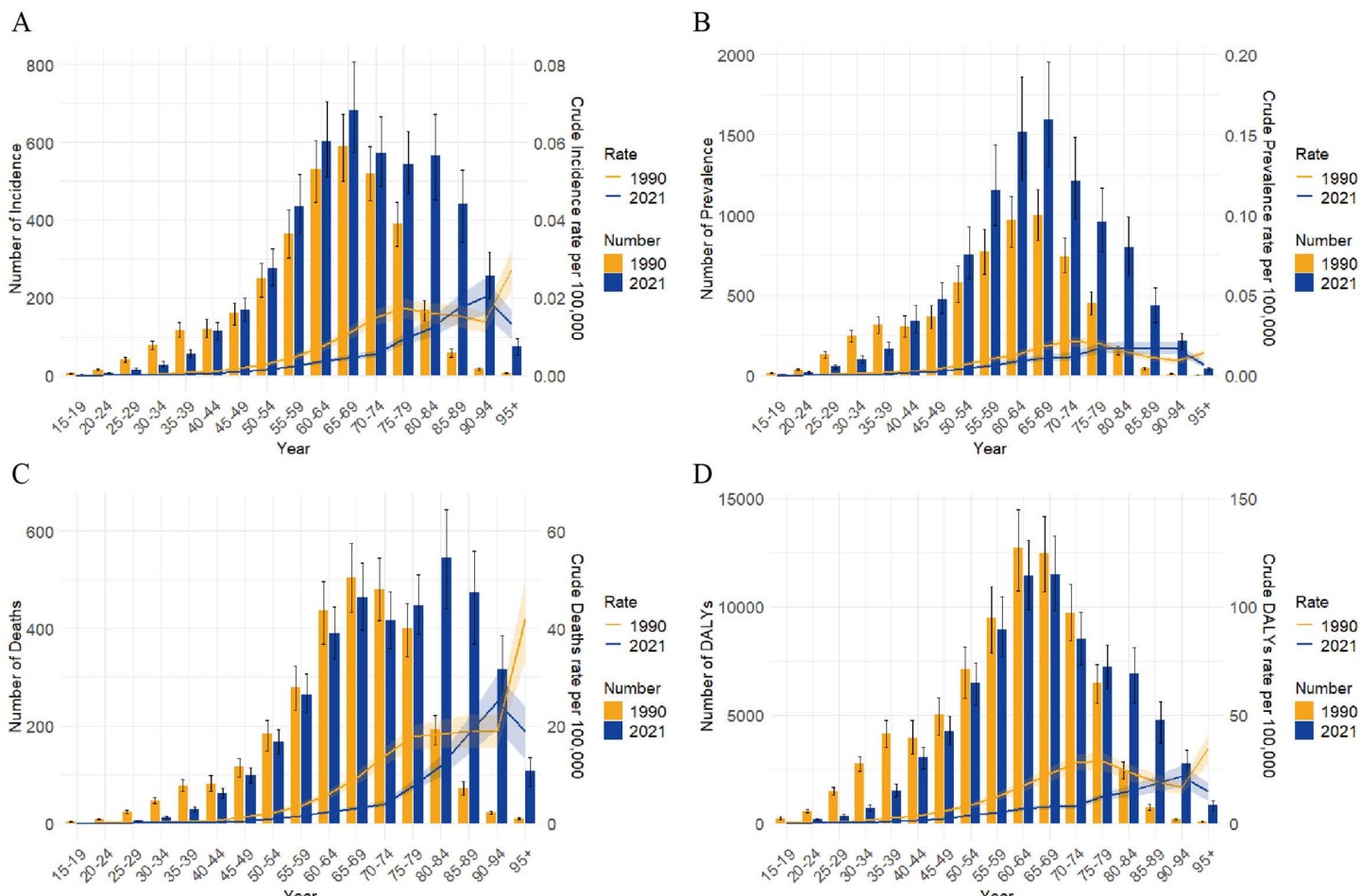

**Fig 2. Comparison of the burden of gastric cancer by age in Taiwan between 1990 and 2021.** (A) Number of incident cases and CIR between 1990 and 2021. (B) Number of prevalent cases and CPR between 1990 and 2021. (C) Number of deaths and CMR between 1990 and 2021. (D)Number of DALYs and CDR between 1990 and 2021. The bar graphs represent the counts, and the line graphs represent the crude rates. Abbreviations: CIR, Crude Incidence Rate; CMR, Crude Mortality Rate; CPR, Crude Prevalence Rate; CDR, Crude DALYs Rate; DALYs, Disability-Adjusted Life Years.

### Age-period-cohort effects on the incidence and mortality rates of gastric cancer in Taiwan

An age-period-cohort model was employed to further assess how age, period, and cohort influence gastric cancer incidence and mortality in Taiwan. The study found that incidence and mortality generally rose with increasing age, for the same age group, rates differed across periods, with 1997–2001 showing higher incidence and mortality than other periods (Fig 5A, 5D). Fig 5B and 5E describe variations in incidence and mortality by birth cohort and age group, noting that older age groups and earlier birth cohorts had higher rates (Fig 5B, 5E). Fig 5C and 5F present period- and cohort-specific differences, similarly indicating higher rates in earlier birth cohorts, with the highest rates within each cohort occurring in 2017–2021 (Fig 5C, 5F).

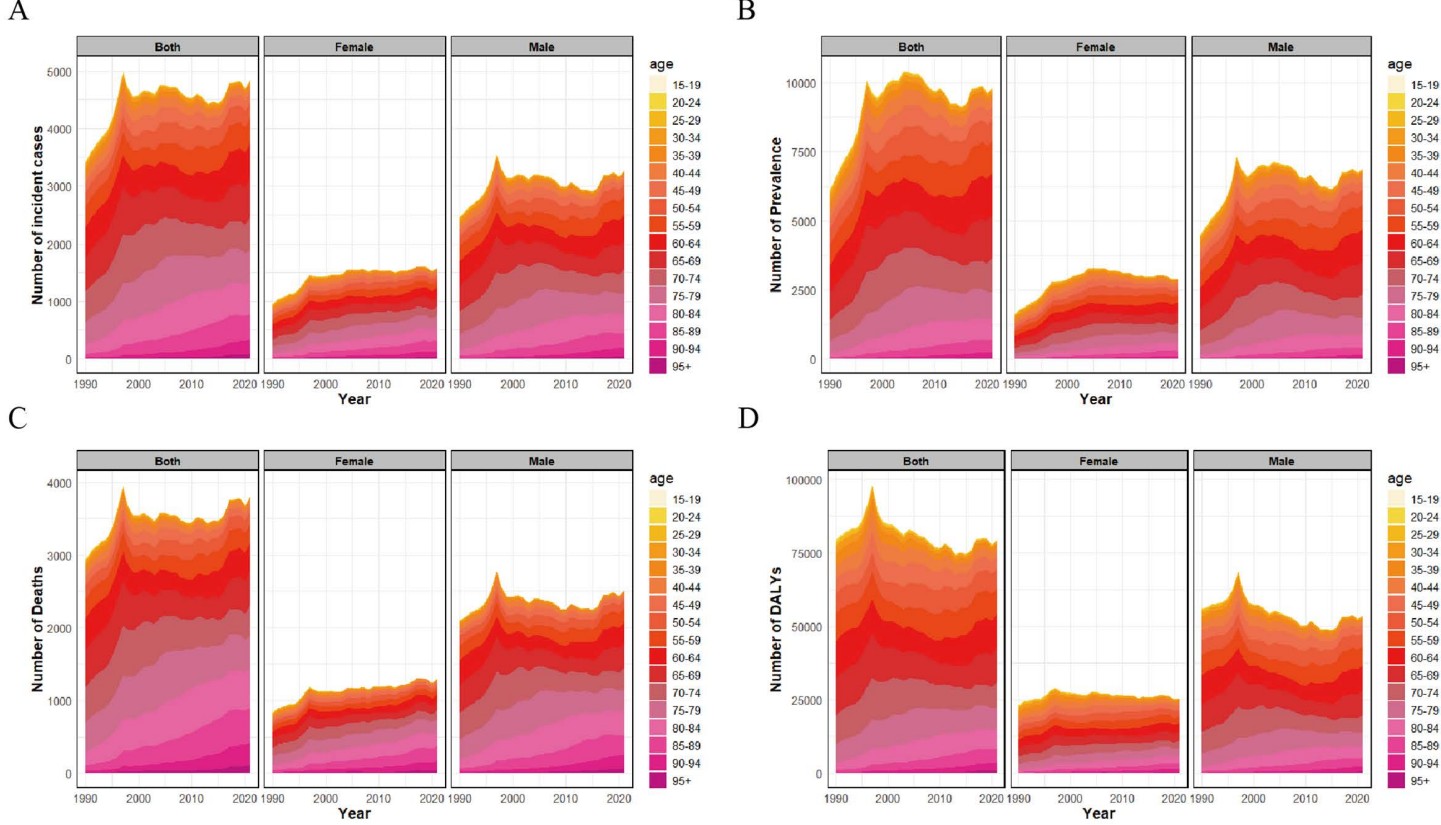

**Fig 3. Trends of age-specific burden of gastric cancer in Taiwan by gender from 1990 to 2021.** (A) Number of incident cases by gender from 1990 to 2021. (B) Number of prevalent cases by gender from 1990 to 2021. (C) Number of deaths by gender from 1990 to 2021. (D) Number of DALYs by gender from 1990 to 2021. Abbreviations: DALYs, Disability-Adjusted Life Years.

## Decomposition analysis of gastric cancer burden in Taiwan

Decomposition analysis identified population aging and growth as drivers of gastric cancer incidence in Taiwan, while epidemiological shifts acted as the primary mitigating factor. Stratification by sex confirmed the same pattern in both males and females (Fig 6A). For prevalence, aging and population growth were the main drivers, with population growth contributing more than aging (Fig 6B). Population growth and aging drove mortality, with aging contributing substantially more than population growth, epidemiological shifts remained the key mitigating factor (Fig 6C). The study found that gastric cancer DALYs in Taiwan decreased from 1990 to 2021, with epidemiological shifts accounting for the largest contribution as the primary mitigating factor, counteracting the effects of aging and population growth (Fig 6D).

## Forecasting trends in gastric cancer incidence and mortality in Taiwan

Using a BAPC model with sex stratification, we forecast gastric cancer incidence and mortality trends in Taiwan over the next 15 years. The ASIR of gastric cancer in Taiwan is expected to decline between 2021 and 2036 (Fig 7A). By 2036, the ASIR will decrease to 12.94 per 100,000 in males, 4.17 per 100,000 in females, and 8.39 per 100,000 in both sexes (S3 Table). The ASMR will similarly decline (Fig 7B), with rates of 9.77 per 100,000 in males, 3.28 per 100,000 in females, and 6.09 per 100,000 in both sexes by 2036 (S3 Table). Incident cases are projected to show a broadly upward trend over the next 15 years (Fig 7A). Male incident cases will rise from 3,264 in 2021 to 5,098 in 2036 (a 56.19% increase), both sexes

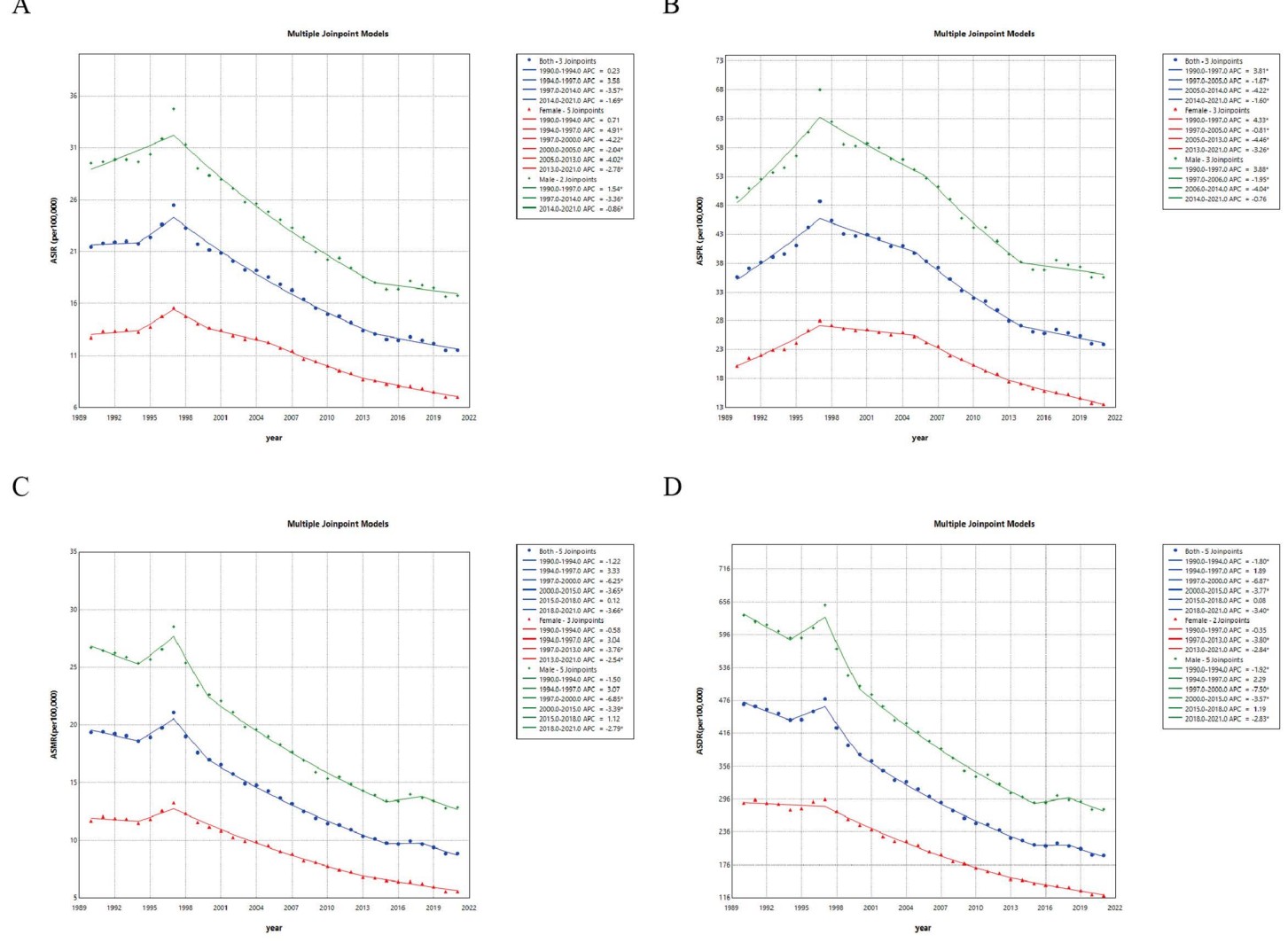

**Fig 4. Joinpoint Analysis of ASIR, ASPR, ASMR, and ASDR of gastric cancer in Taiwan, 1990–2021.** (A) Trends in ASIR of gastric cancer in Taiwan by sex. (B) Trends in ASPR of gastric cancer in Taiwan by sex. (C) Trends in ASMR of gastric cancer in Taiwan by sex. (D) Trends in ASDR of gastric cancer in Taiwan by sex. This analysis included both sexes (blue line), females (red line), and males (green line). Asterisks (*) indicate P < 0.05, suggesting statistical significance. Abbreviations: ASIR, Age-standardized incidence rate; ASPR, Age-standardized prevalence rate; ASMR, Age-standardized mortality rate; ASDR, Age-standardized DALYs rate; DALYs, Disability-Adjusted Life Years.

incident cases will increase from 4,839 in 2021 to 6,603 in 2036 (a 36.45% rise). Female incident cases will decrease slightly, from 1,575 in 2021 to 1,505 in 2036 (4.44% decline; S3 Table). Mortality numbers are also forecast to show a broadly upward trend over the next 15 years (Fig 7B). By 2036, male deaths are predicted to reach 3,478, female deaths 1,294, and both sexes deaths 4,772 (S3 Table).

## Discussion

Based on the GBD 2021 database, this study conducted a comprehensive analysis of the gastric cancer burden in Taiwan between 1990 and 2021. Although ASIR, ASPR, ASMR, and ASDR showed declines, the counts of incident cases, prevalent cases, deaths, and DALYs rose. An epidemiological study using the Taiwan Cancer Registry showed that, in 2021,

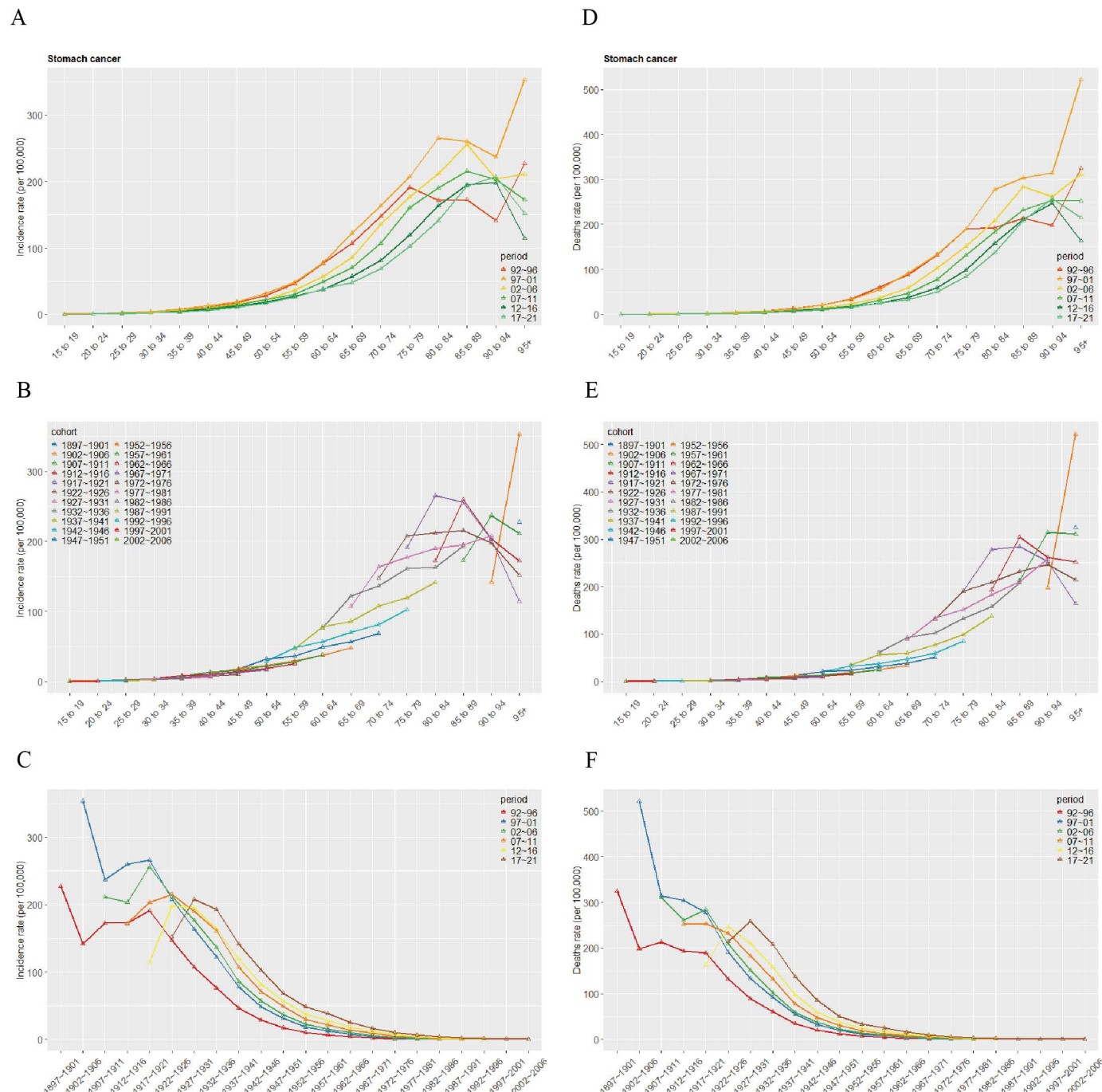

**Fig 5. Age-Period-Cohort Model Analysis of Gastric Cancer Incidence and Mortality in Taiwan.** (A) Age-specific changes in incidence. (B) Incidence changes by cohort. (C) Incidence changes across different periods. (D) Age-specific changes in Death. (E) Mortality changes by cohort. (F) Mortality changes across different periods.

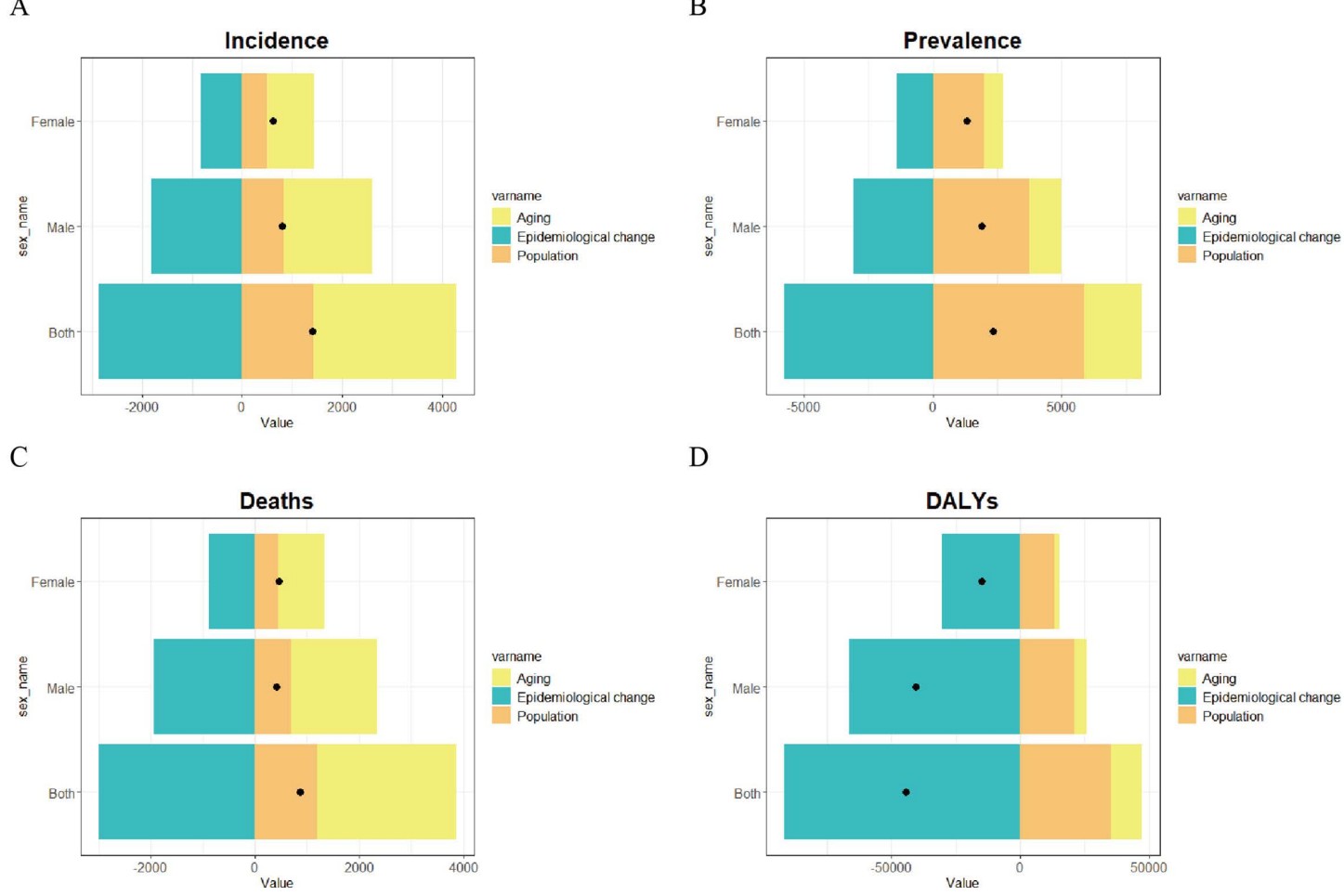

**Fig 6. Decomposition analysis of gastric cancer burden in Taiwan, stratified by gender.** (A) Incidence. (B) Prevalence. (C) Mortality. (D) DALYs. Positive values on the X-axis indicate an increase, negative values indicate a decrease and black dots represent the sum of the contributions of the three factors. Abbreviations: DALYs, Disability-Adjusted Life Years.

gastric cancer in Taiwan had an ASMR of 6.22 per 100,000 and an ASDR of 131.01 per 100,000, values slightly different from those in our study [34]. This variation stems from differing data sources, the Taiwan Cancer Registry is based on hospital registry data in Taiwan, while cancer burden estimates in the GBD study derive from cancer registries, vital statistics, and autopsy data, among other sources. Our research revealed that population aging contributed to increases in gastric cancer incident cases, deaths, and DALYs in Taiwan, whereas epidemiological interventions were the key factor slowing disease progression. This is evident in shifts in population age structure, improved medical conditions, and the implementation of public health care systems, findings that further demonstrate progress in gastric cancer prevention, control, and treatment in Taiwan [35]. Gastric cancer incidence in males in Taiwan has consistently been significantly higher than in females, a sex difference also observed in gastric cancer incidence in other regions [13]. The development and progression of gastric cancer involve multiple risk factors, including lifestyle habits such as smoking, alcohol consumption, and high-sodium diets [7]. The number of male smokers is significantly higher than that of female smokers, and tobacco contains multiple carcinogens that may cause digestive tract ulcers and inflammation, further promoting gastric cancer development [36,37]. Smoking is estimated to contribute to a 4% risk of gastric cancer in females and 11% in

A

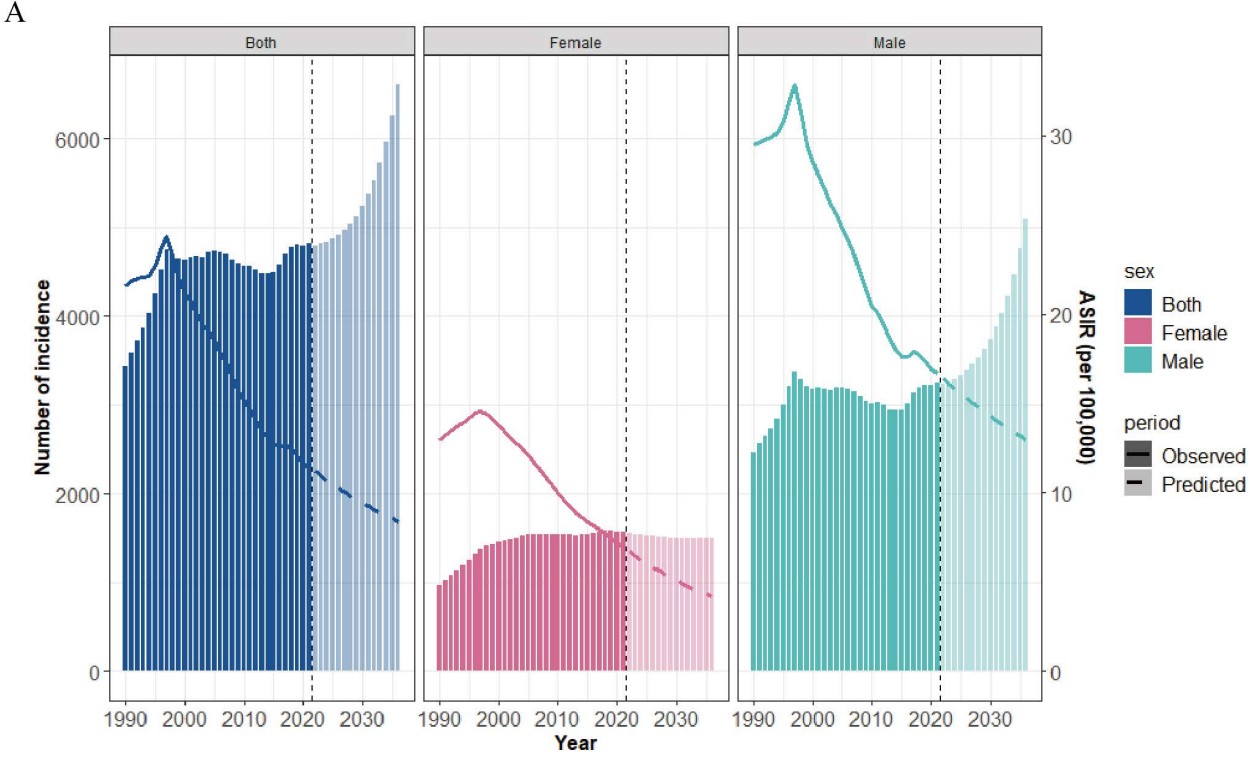

B

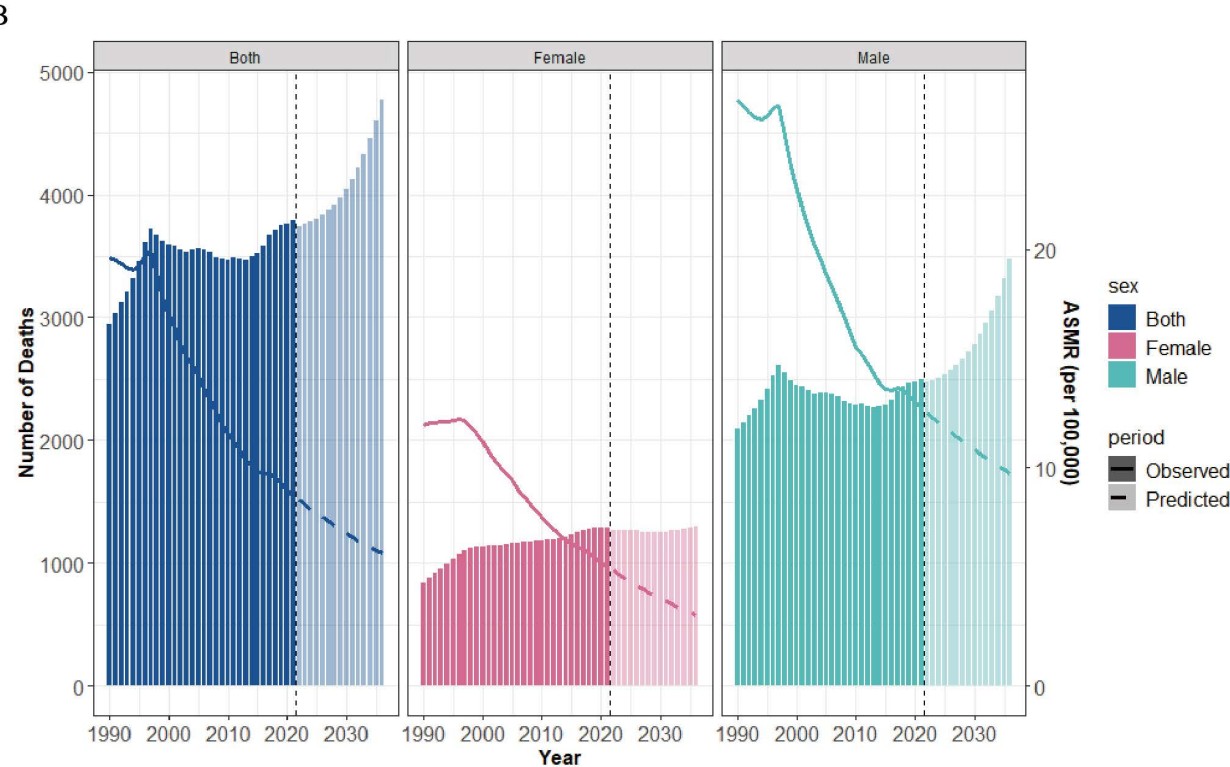

**Fig 7. Projections of Gastric Cancer Incidence and Mortality Burden in Taiwan Through 2036.** (A) ASIR and number of incident cases. (B) ASMR and number of deaths. The solid lines represent the observed ASIR and ASMR, while the dotted lines denote the ASIR and ASMR predicted by the BAPC model. The dark-colored columns indicate the actual number of incident cases and deaths, and the light-colored columns represent the number of incident cases and deaths predicted by the BAPC model. Abbreviations: ASIR, Age-standardized incidence rate; ASMR, Age-standardized mortality rate.

males in developing countries [38,39]. A causal relationship between smoking and gastric cancer has been established in prior studies [15]. Furthermore, research has revealed a significant increase in gastric cancer incidence among asbestos-exposed workers in Taiwan, where male workers far outnumber female workers, a discrepancy that may partly account for the higher gastric cancer incidence in males relative to females in Taiwan [40]. These gender disparities suggest that future public health strategies for gastric cancer prevention and control in Taiwan should incorporate gender-tailored policies to mitigate harm from these risk factors effectively.

Joinpoint analysis found that ASRs of gastric cancer in Taiwan have gone down since 1997. After Taiwan passed the Tobacco Hazards Prevention Act in 1997, fewer people smoked. This may be one reason gastric cancer incidence in Taiwan dropped after 1997 [41]. HP infection can lead to chronic gastritis, which is the main early factor leading to gastric cancer [42]. After 1997, Taiwan started focusing on curing HP infections. As this effort was carried out, HP infections in Taiwan decreased each year, and the burden of gastric cancer went down too [43,44]. Taiwan implemented national health insurance in 1995, a program ensuring access to quality-assured medical services, comprehensive benefits, and convenient treatment [45]. This initiative helped raise public health awareness, enabling more timely diagnosis and treatment of gastric cancer and effectively reducing its disease burden. In 2003, Taiwan introduced the Cancer Prevention and Control Act, which further promoted the prevention and treatment of malignant tumors, thereby contributing to reduced gastric cancer incidence and mortality [35]. Thanks to the implementation of these policies, gastric cancer in Taiwan has been controlled. Future efforts should focus on policy refinement and enhanced enforcement to further reduce the gastric cancer burden.

The age-period-cohort model revealed that gastric cancer incidence and mortality in Taiwan increased with age, consistent with the developmental pattern of most malignancies. Additionally, incidence was higher in 1997–2001 than in other periods; a study reporting trends in multiple cancers in Taiwan found that gastric cancer incidence in both sexes was higher in 1998–2002 than in other periods, which is generally consistent with our results [35]. This phenomenon may be explained by the fact that although Taiwan began emphasizing HP eradication in 1997, increased healthcare access due to economic development and the implementation of the National Health Insurance program led to higher gastric cancer detection rates in 1997–2001 compared with other periods.

The gastric cancer burden among elderly individuals in Taiwan has risen steadily over time. Our decomposition analysis showed that population aging contributed positively to the increasing number of gastric cancer cases and deaths in Taiwan. Population aging is a universal demographic transition globally, driven by economic development and advances in healthcare. Specifically, the aging rate in Taiwan is more than double that of European nations and the United States [46]. Cancer is an age-associated disease, with incidence rising as age increases [47]. Population aging is accompanied by cellular senescence, which is linked to numerous diseases and increases the risk of cancer, cardiovascular conditions, infections, and neurodegenerative diseases [48,49]. Research on global and regional cancer trends in older populations demonstrated that the proportion of total cancer cases in older adults globally increased from 48.6% in 1990 to 56.4% in 2019, while cancer deaths in this group rose from 3 million to 6.2 million [50]. Furthermore, population aging extends individual exposure to tobacco, alcohol, and environmental pollutants, potentially driving up gastric cancer incidence and mortality in Taiwan—findings that align with those from studies on gastric cancer in mainland China [51]. Therefore, health policy development in Taiwan should prioritize the occurrence and prevention of gastric cancer in older populations, including regular gastroscopy screening for elderly individuals and strengthened dietary health education. Research revealed that epidemiological shifts have had a marked alleviating impact on gastric cancer in Taiwan. This pattern is likely explained by the prevention of gastric cancer through lifestyle and dietary modifications, HP eradication, and other strategies, supported by economic growth and policy refinements. Concurrently, progress in gastric cancer treatment—including optimized surgical techniques and the development of novel antineoplastic agents such as immunosuppressants and targeted therapies, has significantly improved outcomes for patients with gastric cancer [52].

Using the BAPC model, we forecast that gastric cancer ASIR and ASMR in Taiwan will decrease over the next 15 years, whereas the number of new cases and deaths will generally increase. A study analyzing gastric cancer epidemic

trends in mainland China reported projections for the next 25 years, rising incident cases and falling ASIR, consistent with our results [18]. The ongoing increase in gastric cancer incidence and mortality in Taiwan will present a substantial threat to public health. Policymakers must implement targeted interventions to alleviate the gastric cancer burden. Priority should be given to precise management of risk factors, sustained advocacy for smoking cessation and alcohol moderation, reduced high-salt food consumption, and greater intake of fresh vegetables and fruits. For high-risk groups (elderly populations), dissemination of gastric cancer prevention and treatment knowledge is critical, alongside emphasizing the value of gastroscopic screening and implementing community-based gastroscopic screening programs to achieve early detection and intervention. As physical function declines with age, older adults often cannot tolerate aggressive treatments. Additionally, most clinical trials don't include elderly participants, so they miss out on the latest treatments. In the future, we should broaden trial inclusion standards, boost teamwork across medical specialties, and improve outcomes for older adults with gastric cancer. Regarding medical resource distribution, there is a need to optimize resource allocation for gastric cancer diagnosis and treatment, with increased investment in primary care facilities to upgrade their gastroscopic equipment and technical proficiency. A telemedicine consultation network should be developed to ensure patients in remote regions can access timely expert diagnoses, preventing treatment delays caused by geographical barriers.

Our study has several limitations. First, variations in the accuracy and completeness of data collection in the GBD database may affect the reliability of our results [21]. Second, due to data constraints, we were unable to further stratify gastric cancer by anatomical site or pathological type, existing studies have identified distinct epidemiological trends between cardia and non-cardia gastric cancer [53]. Third, the GBD model employs smoothing techniques for data processing, which may underestimate annual fluctuations and short-term trend changes in certain diseases. Fourth, biases may arise when extrapolating future gastric cancer burden in Taiwan using the BAPC model. For example, the limited period of data may lead to inaccurate capture of long-term trends by the model, and changes in external factors such as dietary habits have not yet fully manifested their impact on gastric cancer incidence. Additionally, model assumptions are limited which interactions between different age groups (e.g., transmission of dietary habits within families) may introduce biases into predicted results. Uncertainties in external dynamic factors including changes in dietary habits, socioeconomic development, and advances in medical technology also contribute to biases in future disease burden projections.

## Conclusions

This study comprehensively analyzed the burden of gastric cancer in Taiwan, China. Benefiting from the implementation of relevant policies in Taiwan, such as the Tobacco Hazards Prevention Act and the Cancer Prevention and Control Act, along with changes in lifestyle and dietary habits and the eradication of HP, the ASIR and ASMR of gastric cancer have shown a downward trend. However, due to population aging, the number of new gastric cancer cases in Taiwan will continue to rise over the next 15 years. Therefore, detailed prevention and control strategies tailored to different age groups and genders are needed in the future. Additionally, efforts to promote smoking cessation, alcohol restriction, reduced high-salt diets, increased intake of fresh vegetables and fruits, and HP eradication should be sustained across the territory.

## Supporting information

**S1 Table. Joinpoint analysis of the changing trends of ASIR and ASPR of gastric cancer in Taiwan from 1990 to 2021.**
(DOCX)

**S2 Table. Joinpoint analysis of the changing trends of ASMR and ASDR of gastric cancer in Taiwan from 1990 to 2021.**
(DOCX)

**S3 Table. Projections of gastric cancer Incident Cases, Deaths, ASIR, and ASMR in Taiwan till 2036.**
(DOCX)

## Author contributions

**Conceptualization:** Shaoxing Chen, Yadong Lai, Fenglin Chen.

**Data curation:** Canmei Zhong.

**Formal analysis:** Xiaohuang Yang, Shaoxing Chen, Yadong Lai.

**Funding acquisition:** Canmei Zhong.

**Investigation:** Shaoxing Chen.

**Methodology:** Canmei Zhong.

**Project administration:** Yadong Lai, Fenglin Chen.

**Resources:** Canmei Zhong.

**Software:** Xiaohuang Yang.

**Supervision:** Fenglin Chen.

**Validation:** Shaoxing Chen.

**Visualization:** Xiaohuang Yang, Shaoxing Chen.

**Writing – original draft:** Xiaohuang Yang, Shaoxing Chen.

**Writing – review & editing:** Yadong Lai, Fenglin Chen.

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
