## [Decision Letter · Decision Letter 0]

9 Jun 2025

Dear Dr. Yang,

Thank you for submitting your manuscript to PLOS ONE. After careful consideration, we feel that it has merit but does not fully meet PLOS ONE’s publication criteria as it currently stands. Therefore, we invite you to submit a revised version of the manuscript that addresses the points raised during the review process.

We look forward to receiving your revised manuscript.

Kind regards,

Xing-Xiong An, M.D.

Academic Editor

PLOS ONE

Journal Requirements:

This work was supported by grants to the Young and Middle-aged Talents Training Project of Fujian Provincial Health Technology Project (2021GGA021); Startup Fund for Scientific Research, Fujian Medical University (2020QH1098); Joint Funds for the National Key Clinical Specialty Construction Projects of Fujian Province, China (Grant No. 2023 − 1594) and Minimally Invasive Medical Center of Fujian, China (Grant No. [2017]171)

4. Thank you for uploading your study's underlying data set. Unfortunately, the repository you have noted in your Data Availability statement does not qualify as an acceptable data repository according to PLOS's standards.

6. Please include a new copy of Tables 1 and 2 in your manuscript; the current table is difficult to read. Please follow the link for more information: https://blogs.plos.org/plos/2019/06/looking-good-tips-for-creating-your-plos-figures-graphics/

7. We notice that your supplementary table is included in the manuscript file. Please remove them and upload them with the file type 'Supporting Information'. Please ensure that each Supporting Information file has a legend listed in the manuscript after the references list.

Reviewers' comments:

Reviewer's Responses to Questions

**Comments to the Author**

1. Is the manuscript technically sound, and do the data support the conclusions?

Reviewer #1: Yes

Reviewer #2: Yes

Reviewer #3: Yes

2. Has the statistical analysis been performed appropriately and rigorously?

Reviewer #1: No

Reviewer #2: Yes

Reviewer #3: Yes

3. Have the authors made all data underlying the findings in their manuscript fully available?

Reviewer #1: Yes

Reviewer #2: Yes

Reviewer #3: Yes

4. Is the manuscript presented in an intelligible fashion and written in standard English?

Reviewer #1: Yes

Reviewer #2: Yes

Reviewer #3: No

Reviewer #1: Thank you for the opportunity to review this manuscript, which examines the long-term trends in the gastric cancer burden in Taiwan from 1990 to 2021, based on the GBD 2021 database, and projects outcomes until 2036. The study design is generally robust, the methodology is appropriate for the research aims, and the findings hold valuable implications for public health policymaking. Nevertheless, I have several specific points that merit further clarification or revision to strengthen the manuscript:

1. It would be helpful to clarify whether the BAPC model takes into account dynamic factors that may influence the future burden of gastric cancer, such as changes in dietary habits. Please specify the assumptions underlying the BAPC model and discuss potential sources of bias in extrapolating the results.

2. While the manuscript references an APC model, it does not detail how the inherent collinearity among age, period, and cohort variables was handled. Providing additional methodological details on how you addressed linear dependence will bolster the credibility of your findings.

3. The study predicts a decrease in both the ASIR and ASMR for gastric cancer in Taiwan by 2036. However, despite the overall decline in these rates, the number of incident cases is projected to rise, particularly due to Taiwan's aging population. Given this trend, I recommend expanding the discussion to include tailored interventions for this demographic. This would make the findings more actionable and relevant for public health policymakers.

4. On Line 167, the term “AMPR” appears, which should presumably be “ASMR”.

Reviewer #2: The conclusion of The Trend analysis and projection of the gastric cancer disease burden in Taiwan during 1990 - 2021: An analysis of the global burden of disease study is stable. The figures are needed to revised. There are some grammatical errors to corrected.

Reviewer #3: It is my pleasure to review the manuscript by Yang et al. The authors have posed a meaningful research question and provided well-supported answers through the analysis of public datasets using established epidemiological methods. I have the following comments and recommendations for the authors to further improve the manuscript.

1. Lines 68 – 72. Please add the following two references to strengthen the statement. Lee YC et al. Screening for Helicobacter pylori to Prevent Gastric Cancer: A Pragmatic Randomized Clinical Trial. JAMA 2024 Nov 19;332(19):1642-1651. Liou JM et al. Screening and eradication of Helicobacter pylori for gastric cancer prevention: the Taipei global consensus. Gut 2020 Dec;69(12):2093-2112.

2. According to the study authored by Liu PC et al. Disease Burden of 30 Cancer Groups in Taiwan from 2000 to 2021. J Epidemiol Glob Health 2025 Apr 22;15(1):62. Regarding stomach cancer, the age-standardized rate for mortality in 2000 was estimated to be 13.40 per 100,000 population for both sexes. Twenty-one years later in 2021, it was 6.22/100,000. It would be much appreciated if the authors can cite and discuss the subtle differences between the current study and that from the Liu’s study.

3. Table 1: Please remove ‘(Province of China)’ following Taiwan to avoid introducing political or ideological implications into the study. We also recommend removing similar references throughout the manuscript to maintain neutrality and ensure appropriateness for an international readership. The term ‘mainland China’ may be retained to reflect goodwill and provide clarity where relevant. We believe that such adjustments may encourage broader acceptance and citation of your work among Taiwanese scholars. A win-win strategy.

4. Line 520: please modify the word, ‘province’ to territory, or simply island.

5. Do the authors have the capacity to perform a health disparities analysis to examine why one population (mainland China) bears a higher disease burden than another (Taiwan), by disentangling the effects of demographic composition, disease risk, and other contributing factors?

6. Line 388, a reference is needed following the statement.

**Do you want your identity to be public for this peer review?** For information about this choice, including consent withdrawal, please see our Privacy Policy

Reviewer #1: No

Reviewer #2: No

Reviewer #3: No

---

## [Author Response · Author response to Decision Letter 1]

22 Jul 2025

List of responses

Dear Editors and Reviewers:

Thank you for your letter and for the reviewers’ comments concerning our manuscript entitled “Trend analysis and projection of the gastric cancer disease burden in Taiwan during 1990 - 2021: An analysis of the global burden of disease study 2021” (Manuscript Number: PONE-D-25-03556). Those comments are all valuable and very helpful for revising and improving our paper, as well as the important guiding significance for our research. We have revised the manuscript, according to the comments and suggestions of reviewers and the editor, and responded, point by point, to the comments as listed below, which we hope will meet with approval. Revised portions are marked in red in the paper. The main corrections in the paper and the responses to the reviewer’s comments are as follows:

Response to Journal Requirements

The authors’ answer: We have carefully reviewed PLOS ONE's style requirements and ensured that our revised manuscript adheres to all specified guidelines, including file naming conventions and formatting standards.

2. Thank you for stating the following financial disclosure: This work was supported by grants to the Young and Middle-aged Talents Training Project of Fujian Provincial Health Technology Project (2021GGA021); Startup Fund for Scientific Research, Fujian Medical University (2020QH1098); Joint Funds for the National Key Clinical Specialty Construction Projects of Fujian Province, China (Grant No. 2023 − 1594) and Minimally Invasive Medical Center of Fujian, China (Grant No. [2017]171).Please state what role the funders took in the study. If the funders had no role, please state: "The funders had no role in study design, data collection and analysis, decision to publish, or preparation of the manuscript." If this statement is not correct you must amend it as needed. Please include this amended Role of Funder statement in your cover letter; we will change the online submission form on your behalf.

The authors’ answer: Thank you for your guidance on the funder role statement. We confirm the following regarding the involvement of the funding agencies in this study: the Young and Middle-aged Talents Training Project of Fujian Provincial Health Technology Project (2021GGA021), which played a role in the decision to publish. Additionally, the Startup Fund for Scientific Research Fujian Medical University (2020QH1098), Joint Funds for the National Key Clinical Specialty Construction Projects of Fujian Province, China (Grant No. 2023 − 1594) and Minimally Invasive Medical Center of Fujian, China (Grant No. [2017]171) provided support for this work, contributing to data collection and analysis.

The authors’ answer: We have carefully revised the manuscript to include the full ethics statement in the Methods section, as specified. Details are as follows: All data in this study were sourced from the publicly accessible GBD database and subjected to secondary analysis, rendering ethical approval and informed consent unnecessary (lines 149-152).

4. Thank you for uploading your study's underlying data set. Unfortunately, the repository you have noted in your Data Availability statement does not qualify as an acceptable data repository according to PLOS's standards. At this time, please upload the minimal data set necessary to replicate your study's findings to a stable, public repository (such as figshare or Dryad) and provide us with the relevant URLs, DOIs, or accession numbers that may be used to access these data. For a list of recommended repositories and additional information on PLOS standards for data deposition, please see https://journals.plos.org/plosone/s/recommended-repositories.

The authors’ answer: Thank you for your feedback regarding the data availability. We appreciate the opportunity to clarify the data deposition for our study. The underlying data used in this study were obtained from the publicly accessible Global Burden of Disease 2021 (GBD 2021) database, which is a stable, public repository meeting PLOS's standards for data deposition. The data can be accessed via the following URL: http://ghdx.healthdata.org/gbd-results-tool.We have updated the Data Availability statement in the revised manuscript to explicitly reference the GBD 2021 database and its URL. The specific steps for downloading the data are as follows: for the cause, “Stomach cancer”; for measurements, “incidence, prevalence, deaths and DALYs”; for location, “China and Taiwan (Province of China)”; for years, “1990–2021”; for metrics, “number, percent, and rate”; for sex, “both, male and female”. Age stratification included: <5 years,5-9 years,10-14 years,15-19 years,20-24 years,25-29 years,30-34 years,35-39 years,40-44 years,45-49 years,50-54 years,55-59 years,60-64 years,65-69 years,70-74 years,75-79 years,80-84 years,85-89 years,90-94 years,95+years, All ages, Age-standardized.

The authors’ answer: Thank you for your guidance on the data availability policy. We appreciate the opportunity to clarify our data sharing plan and address the requirements promptly. The study utilizes publicly available data from the Global Burden of Disease 2021 (GBD 2021) database, which is accessible via http://ghdx.healthdata.org/gbd-results-tool. As this data is already freely available, no access restrictions apply, and it fully complies with PLOS's open data policy. We have amended the data availability statement to clarify that the data sharing protocol will be determined before manuscript acceptance.

6. Please include a new copy of Tables 1 and 2 in your manuscript; the current table is difficult to read. Please follow the link for more information:

https://blogs.plos.org/plos/2019/06/looking-good-tips-for-creating-your-plos-figures-graphics/

The authors’ answer: Thank you for the feedback on the readability of Tables 1 and 2. We have revised the table formats to enhance clarity, by the PLOS guidelines for figure and table presentation. The revised Table 1 has been inserted into the corresponding section of the manuscript (line 570), and Table 2 has been moved to the supplementary materials as Supplementary Tables 1 and 2.

7. We notice that your supplementary table is included in the manuscript file. Please remove them and upload them with the file type 'Supporting Information'. Please ensure that each Supporting Information file has a legend listed in the manuscript after the references list.

The authors’ answer: Thank you for your meticulous guidance on the supplementary materials. In the revised manuscript, we have revised the supplementary tables as required to ensure full compliance with the submission guidelines of PLOS ONE.

Response to Reviewer 1

Reviewer #1 Comment 1: It would be helpful to clarify whether the BAPC model takes into account dynamic factors that may influence the future burden of gastric cancer, such as changes in dietary habits. Please specify the assumptions underlying the BAPC model and discuss potential sources of bias in extrapolating the results.

The authors’ answer: Thank you for your valuable suggestion. The Bayesian Age-Period-Cohort (BAPC) model correlates in predicting future disease burden trends by analyzing age, period, and cohort effects in historical data. For model construction, we primarily analyzed existing epidemiological data and historical trends from the Global Burden of Disease (GBD) database. While the database includes partial dietary risk factors for gastric cancer (e.g., high-sodium diets), it lacks risk data on other dietary habits. We also recognize that changes in dynamic factors such as dietary habits are complex and may be influenced by the combined effects of socioeconomic, cultural, and health awareness factors, which affect model predictions to some extent. The BAPC model is based on the following assumptions: 1) Independence assumption: Effects between different ages, periods, and cohorts are assumed to be mutually independent within the model. 2) Smoothness assumption: Changes in age, period, and cohort effects are assumed to be smooth rather than abrupt. 3) Data representativeness assumption: The data used are considered representative of the overall study population. We summarized these assumptions in the methods section and presented the underlying equations (lines 136–145). Potential sources of bias may arise when extrapolating BAPC model results to the future: For example, the limited period of data may lead to inaccurate capture of long-term trends by the model, and changes in external factors such as dietary habits have not yet fully manifested their impact on gastric cancer incidence. Additionally, model assumptions are limited: interactions between different age groups (e.g., transmission of dietary habits within families) may introduce biases into predicted results. Uncertainties in external dynamic factors—including changes in dietary habits, socioeconomic development, and advances in medical technology—also contribute to biases in future disease burden projections. We have addressed in the limitations section (lines 412–420).

Reviewer #1 Comment 2: While the manuscript references an APC model, it does not detail how the inherent collinearity among age, period, and cohort variables was handled. Providing additional methodological details on how you addressed linear dependence will bolster the credibility of your findings.

The authors’ answer: We appreciate the reviewer’s valuable suggestion. The inherent collinearity between age, period, and cohort (APC) variables is of substantial importance and represents a key component that demands detailed clarification in our research. These three variables exhibit a linear dependence—specifically, "period = age + cohort"—which introduces instability in model parameter estimation and complicates interpretation. In the present study, we utilized the intrinsic estimator to address collinearity by applying linear constraints to the parameter space, yielding unique, stable estimates with minimal bias. Given the specific characteristics of the intrinsic estimator, we stratified ages into 17 groups (15–19 years to ≥95 years) at 5-year intervals, alongside 22 birth cohorts (1897–1901 to 2002–2006) and 6 periods (1992–1996 to 2017–2021), to present gastric cancer incidence and mortality for specific ages, periods, and birth cohorts. Details are provided in the "Age–Period–Cohort Analysis" section of the Methods.

Reviewer #1 Comment 3: The study predicts a decrease in both the ASIR and ASMR for gastric cancer in Taiwan by 2036. However, despite the overall decline in these rates, the number of incident cases is projected to rise, particularly due to Taiwan's aging population. Given this trend, I recommend expanding the discussion to include tailored interventions for this demographic. This would make the findings more actionable and relevant for public health policymakers.

The authors’ answer: Thank you for this insightful suggestion. In the revised "Discussion" section, we added content outlining targeted interventions, including precise management of risk factors; enhanced gastroscopic screening, early intervention, and multidisciplinary collaboration for high-risk populations (elderly individuals); development of personalized treatment plans; and increased resource investment in primary care, alongside upgrades to gastroscopic equipment and technical capabilities.

Reviewer #1 Comment 4: On Line 167, the term “AMPR” appears, which should presumably be “ASMR”.

The authors’ answer: We appreciate you identifying this error. You are correct that "AMPR" here is a typo and should be "ASMR." This will be corrected in the revised manuscript (line 163), and we will carefully review the entire text to ensure no similar errors are overlooked.

Response to Reviewer 2

Reviewer #2 Comment: The conclusion of The Trend analysis and projection of the gastric cancer disease burden in Taiwan during 1990 - 2021: An analysis of the global burden of disease study is stable. The figures are needed to revised. There are some grammatical errors to corrected.

The authors’ answer: We appreciate your review of our study and your valuable comments. All figures and tables will undergo careful inspection and revision to ensure they clearly and accurately reflect the study findings. Specifically, minor issues in the legend of Figure 6 have been addressed, with the revised figure reuploaded. Regarding the incomplete presentation of Tables 1 and 2, revisions have been made: the revised Table 1 has been inserted into the manuscript at the corresponding position (line 570), and Table 2 has been moved to the supplementary materials as Supplementary Tables 1 and 2. Additionally, a colleague whose native language is English has proofread and revised the entire manuscript to ensure fluent, accurate language that conforms to English expression conventions.

Response to Reviewer 3

Reviewer #3 Comment 1: Lines 68 – 72. Please add the following two references to strengthen the statement. Lee YC et al. Screening for Helicobacter pylori to Prevent Gastric Cancer: A Pragmatic Randomized Clinical Trial. JAMA 2024 Nov 19;332(19):1642-1651. Liou JM et al. Screening and eradication of Helicobacter pylori for gastric cancer prevention: the Taipei global consensus. Gut 2020 Dec;69(12):2093-2112.

The authors’ answer: We appreciate your valuable suggestions. The two additional references you recommended will significantly strengthen our argument that Helicobacter pylori infection is a risk factor for gastric cancer. These references will be cited in the manuscript (lines 59–63). Thank you again for your input, which will enhance the comprehensiveness and persuasiveness of our study.

Reviewer #3 Comment 2: According to the study authored by Liu PC et al. Disease Burden of 30 Cancer Groups in Taiwan from 2000 to 2021. J Epidemiol Glob Health 2025 Apr 22;15(1):62. Regarding stomach cancer, the age-standardized rate for mortality in 2000 was estimated to be 13.40 per 100,000 population for both sexes. Twenty-one years later in 2021, it was 6.22/100,000. It would be much appreciated if the authors can cite and discuss the subtle differences between the current study and that from the Liu’s study.

The authors’ answer: Thank you for this insightful suggestion. We will cite the study by Liu PC et al. and compare the differences between the two studies in the Discussion section. Specifically, there are minor discrepancies between our study and that of Liu PC et al. regarding the age-standardized mortality rate (ASMR) and age-standardized disability rate (ASDR) of gastric cancer in Taiwan in 2021. This variation stems from differing data sources: the Taiwa

---

## [Decision Letter · Decision Letter 1]

17 Aug 2025

Trend analysis and projection of the gastric cancer disease burden in Taiwan during 1990 - 2021: An analysis of the global burden of disease study 2021

PONE-D-25-03556R1

Dear Dr. Yang,

We’re pleased to inform you that your manuscript has been judged scientifically suitable for publication and will be formally accepted for publication once it meets all outstanding technical requirements.

Kind regards,

Xing-Xiong An, M.D.

Academic Editor

PLOS ONE

Additional Editor Comments (optional):

Thanks for the authors' efforts to comprehensively improve your manuscript according to editor's and reviewers' comments. I am pleased to inform you that your paper can be accepted for publication now.

Reviewers' comments:

Reviewer's Responses to Questions

**Comments to the Author**

Reviewer #2: All comments have been addressed

Reviewer #3: All comments have been addressed

2. Is the manuscript technically sound, and do the data support the conclusions?

Reviewer #2: Yes

Reviewer #3: Yes

3. Has the statistical analysis been performed appropriately and rigorously?

Reviewer #2: Yes

Reviewer #3: Yes

4. Have the authors made all data underlying the findings in their manuscript fully available?

Reviewer #2: Yes

Reviewer #3: Yes

5. Is the manuscript presented in an intelligible fashion and written in standard English?

Reviewer #2: Yes

Reviewer #3: Yes

Reviewer #2: This study comprehensively analyzed the burden of gastric cancer in Taiwan, China. Benefiting from the implementation of relevant policies in Taiwan—such as the Tobacco Hazards Prevention Act and the Cancer Prevention and Control Act—along with changes in lifestyle and dietary habits and the eradication of HP, the ASIR and ASMR of gastric cancer have shown a downward trend. However, due to population aging, the number of new gastric cancer cases in Taiwan will continue to rise over the next 15 years. Therefore, detailed prevention and control strategies tailored to different age groups and genders are needed in the future. Additionally, efforts to promote smoking cessation, alcohol restriction, reduced high-salt diets, increased intake of fresh vegetables and fruits, and HP eradication should be sustained across the territory.

All mentioned above is the conclusion that this manuscript has been drawn.

Since the comments have been addressed , acceptance is that I the recommend.

Reviewer #3: The authors have made a commendable effort in revising the manuscript. 

**Do you want your identity to be public for this peer review?** For information about this choice, including consent withdrawal, please see our Privacy Policy

Reviewer #2: **Yes: ** genlin lu

Reviewer #3: **Yes: ** Victor C. Kok

---

## [Editor Report · Acceptance letter]

PONE-D-25-03556R1

PLOS ONE

Dear Dr. Yang,

I'm pleased to inform you that your manuscript has been deemed suitable for publication in PLOS ONE. Congratulations! Your manuscript is now being handed over to our production team.

Kind regards,

on behalf of

Dr. Xing-Xiong An

Academic Editor

PLOS ONE